# In Search of a Target Gene for a Desirable Phenotype in Aquaculture: Genome Editing of Cyprinidae and Salmonidae Species

**DOI:** 10.3390/genes15060726

**Published:** 2024-06-01

**Authors:** Svetlana Yu. Orlova, Maria N. Ruzina, Olga R. Emelianova, Alexey A. Sergeev, Evgeniya A. Chikurova, Alexei M. Orlov, Nikolai S. Mugue

**Affiliations:** 1Laboratory of Molecular Genetics, Russian Federal Research Institute of Fisheries and Oceanography, 105187 Moscow, Russia; kordicheva@rambler.ru (S.Y.O.);; 2Department of Biological Evolution, Faculty of Biology, Lomonosov Moscow State University, 119234 Moscow, Russia; 3Laboratory of Oceanic Ichthyofauna, Shirshov Institute of Oceanology, Russian Academy of Sciences, 117218 Moscow, Russia; 4Laboratory of Behavior of Lower Vertebrates, Severtsov Institute of Ecology and Evolution, Russian Academy of Sciences, 119071 Moscow, Russia; 5Department of Ichthyology, Dagestan State University, 367000 Makhachkala, Russia; 6Laboratory of Genome Evolution and Speciation, Institute of Developmental Biology Russian Academy of Sciences, 117808 Moscow, Russia

**Keywords:** ZFN, TALEN, CRISPR/Cas9, *Danio rerio*, *Oncorhynchus mykiss*, *Salmo salar*, *Cyprinus carpio*, *Carassius auratus cuvieri*, *Carassius gibelio*

## Abstract

Aquaculture supplies the world food market with a significant amount of valuable protein. Highly productive aquaculture fishes can be derived by utilizing genome-editing methods, and the main problem is to choose a target gene to obtain the desirable phenotype. This paper presents a review of the studies of genome editing for genes controlling body development, growth, pigmentation and sex determination in five key aquaculture Salmonidae and Cyprinidae species, such as rainbow trout (*Onchorhynchus mykiss*), Atlantic salmon (*Salmo salar*), common carp (*Cyprinus carpio*), goldfish (*Carassius auratus*), Gibel carp (*Carassius gibelio*) and the model fish zebrafish (*Danio rerio*). Among the genes studied, the most applicable for aquaculture are *mstnba*, *pomc*, and *acvr2*, the knockout of which leads to enhanced muscle growth; *runx2b*, mutants of which do not form bones in myoseptae; *lepr*, whose lack of function makes fish fast-growing; *fads2*, *Δ6abc/5Mt*, and *Δ6bcMt*, affecting the composition of fatty acids in fish meat; *dnd mettl3*, and *wnt4a*, mutants of which are sterile; and disease-susceptibility genes *prmt7*, *gab3*, *gcJAM-A*, and *cxcr3.2*. Schemes for obtaining common carp populations consisting of only large females are promising for use in aquaculture. The immobilized and uncolored zebrafish line is of interest for laboratory use.

## 1. Introduction

Wild aquatic bioresources are an important source of animal protein and micronutrients worldwide, especially for human populations in coastal regions. From 1961 to 2019, global fisheries’ product consumption increased from 9.0 to 20.5 kg live-weight equivalent per capita. In 2020, this figure declined to 20.2 kg. With ever-increasing demand, total fishery and aquaculture production is increasing. In 1950, it was 19 million tonnes live-weight equivalent; in 2018, it reached a historic high of about 179 million tonnes [1], and in 2020 total fisheries and aquaculture production was 178 million tonnes [2]. Aquaculture is the controlled production of aquatic organisms, an important part of agriculture. In 2020, this sector produced 99 million tonnes, or 56% of the total food production from aquatic animals available for human consumption. In comparison, this proportion was only 4% in 1970 [2]. In 2018, the industry employed about 20.5 million people worldwide [1]; in 2022, it employed 58.5 million people [2]. Due to the COVID-19 pandemic, commercial aquaculture faced changes in consumer demand, market disruptions, and logistical difficulties. The industry is now recovering from the crisis. FAO predicts that due to continued population growth and dietary diversification, there will be an increase in demand for food products. By 2030, total fisheries and aquaculture production is predicted to reach 202 million tonnes, of which 106 million tonnes will come from aquaculture [2].

Aquaculture production can be increased through the genetic improvement of farmed species [3]. Today, highly productive strains of rainbow trout [4], Atlantic salmon [5], Nile tilapia (*Oreochromis niloticus*) [6], and common carp [7] already exist. In the future, new strains and lines can be created using traditional selective breeding, genomic selection methods, as well as biotechnology methods (polyploidization, genetic modification, and genome editing). Genome editing with CRISPR/Cas (Clustered regularly interspaced short palindromic repeats/CRISPR-associated protein) system is highlighted as one of the aquaculture technologies of the future.

For aquaculture, genes that determine economically valuable traits such as growth, development, pigmentation, sex determination, reproductive ability, fatty acid synthesis, and disease resistance can be edited [8,9,10,11,12,13,14,15,16]. These review articles cited the underlying work on editing such genes in a fairly wide range of artificially bred species.

In practical activity, the choice of a gene for editing that determines a desirable phenotype is crucial. It is important to understand which genes’ knockout or modification will lead to the expression of the target trait, and which genes may lead to undesirable effects. The functions of many genes have been studied in the model species zebrafish and several aquaculture species. This review is intended to help researchers navigate the diversity of work on genome editing in fishes. To facilitate the perception of diverse and disparate information on the issues under consideration, we have compiled a table indicating the edited gene, mutant phenotype, and detected gene function for each fish species. In this review, we consider work carried out on key Salmonidae and Cyprinidae fish species, including the model species zebrafish and important aquaculture fish, including rainbow trout, Atlantic salmon, common carp, goldfish, and Gibel carp. In this review, we identify the results of gene editing studies that are the most promising for aquaculture. We also provide general information on the principles of genome editing in fish.

## 2. Genome Editing in Salmonidae and Cyprinidae Aquaculture Fish Species

### 2.1. Initial Studies on the Application of Genome Editing in Fish

Fish genome editing is often used in basic research to decipher the functions of individual genes and elucidate evolutionary mechanisms at the molecular level. Fish, which are evolutionarily related to “higher” vertebrates, provide important model systems for the study of genetic and evolutionary patterns. Effective gene editing requires knowledge of the genome, and the development of high-throughput sequencing technology has facilitated the sequencing of more than two hundred fish genomes [17].

From 2008 to 2013, the ZFN, TALEN, and CRISPR/Cas9 systems were successfully applied to zebrafish [18,19,20]. This marked the first successful gene editing in fish, and the application of ZFN and TALEN on the yellow catfish (*Tachysurus fulvidraco*) (2011) [21] was the first instance of gene editing in an aquaculture fish. In 2014, the CRISPR/Cas9 method was successfully applied to edit the genes of Nile tilapia [22] and Atlantic salmon [23]. Subsequently, gene-editing technology has been successfully applied to other economically important species such as common carp [24], grass carp (*Ctenopharyngodon Idella*) [25], and tongue sole (*Cynoglossus semilaevis*) [26]. These systems allow a wide range of applications, including efficient gene knockout and functional studies of sex, growth, development, and immunity and selection for disease resistance, stress tolerance, and rapid growth. The application of gene editing technology in fish is actively developing and can be used to develop new aquaculture breeds in the long term [13].

Today, CRISPR/Cas9 technology is most often used to edit fish genomes. According to PubMed article database statistics (https://pubmed.ncbi.nlm.nih.gov/, accessed on 28 May 2024), ZFN technology was more frequently used for zebrafish before 2008 and TALEN in 2008–2013, and CRISPR/Cas9 method became dominant after 2014 (Figure 1).

### 2.2. Zebrafish as a Model Object in Studies Using Genome Editing

Over the past forty years, zebrafish has been a widely used model organism for studies of vertebrate development and disease, as well as for the development of molecular genetic techniques. Studies in genetics and experimental embryology have complemented each other, resulting in significant advances in understanding of the regulation of embryo formation, organogenesis, and nervous system development. Research using this model system has expanded into various areas of molecular biology, including the genetic regulation of aging, regeneration, and animal behavior. Zebrafish is a popular experimental model because of the ease with which they can be grown and reproduced, the short generation time, their small size, the low cost, the ability to produce hundreds of embryos daily, the transparency of eggs and larvae, and the speed of development during the early stages of ontogeny. Importantly, uncomplicated protocols can be developed for this system that can be reproduced by laboratory personnel with any level of experience, including students [27,28]. An analysis of articles in the PubMed database showed that the zebrafish is frequently used in genome editing experiments to create models of not only human disease and behavior but also economically valuable traits, with most work focusing on the functions of specific genes.

Zebrafish is valuable as a model system not only because of the convenience of working with it but also because about 70% of human genes have at least one specific ortholog in the genome of this fish species [29]. Existing interspecies differences (e.g., human, mouse, zebrafish) have been extensively investigated, including for mechanisms of nucleic acid repair following genome editing. Cas9-induced mutations are generally considered stochastic and unpredictable, making the method difficult to apply where precise genetic changes are required. However, through a systematic approach and analysis of the results of genome editing of multiple sites in four different species (human and mouse cells, domestic silk moth (*Bombyx mori*) and zebrafish), Cas9-induced mutations were found to be similar in mutation types but significantly different in patterns [30].

In creating disease models, known mutations in genes that cause particular genetic syndromes in humans are obtained, such as muscle laminopathy [31], Charlevoix-Saguenay spastic ataxia [32], Marfan syndrome [33], Sanfilippo syndrome [34], Joubert syndrome [35], Bernard-Soulier syndrome [36], Xia-Gibbs syndrome [37], Lee syndrome [38], Laron syndrome [39], Aicardi-Gutierrez syndrome [40], Finnish-type nephrotic syndrome [41], fragile cornea syndrome [42], multicentric carpotarsal osteolysis syndrome [43], Bietti crystalline dystrophy [44], sarcoglycanopathy [45], autosomal recessive microcephaly [46], sphingolipidosess [47], and mitochondrial diseases caused by *polg* gene mutations [48].

Another approach to modeling is to simulate the symptoms of the target disease through mutations in genes with known function. Individuals and lines have been created that exhibit analogs of human somatic diseases such as cataract [49,50], myopia [51], exfoliative syndrome [52], retinal dysfunction [53,54], congenital heart defect [55], cardiac hypertrophy [56], dilated cardiomyopathy [57], arrhythmia [58], autoinflammatory syndrome [59], metabolic syndrome [60], diabetes and obesity [61], tuberculosis [62], thrombocytopenia [63], pediatric intestinal pseudoobstruction [64], pediatric cirrhosis [65], congenital hypothyroidism [66], fatty or alcoholic hepatosis [67], and scoliosis [68,69]. Genome editing has been used to model human tumors such as liver cancer [70], paraganglioma [71], skin melanoma [72], and epithelioid sarcoma [73]. Among neurologic diseases, amyotrophic lateral sclerosis [74], epilepsy [75,76,77,78,79], Hirschprung’s disease [80], autism [81,82,83,84], spastic paraplegia [85], restless legs syndrome [86], insomnia [87], microcephaly [46], neurotransmitter function of dopamine [88], monoamine [89], and others [90,91]. Cleft lip and cleft palate mutants also have been tested [92]. Edited zebrafish are used to study feeding behavior [93], social behavior [94,95], hyperactivity [96], domestication patterns [97], and circadian rhythms [98]. These models are used to study the genetic basis of disease development and drug testing.

Much of the work has focused on the role of specific genes in organ formation and function, including the lens [99], retina [100,101,102], optic neurons [103,104], ocular vessels [105,106], inner ear [107,108,109,110,111], vestibular apparatus [111], otoliths [112], brain [89,113,114,115,116,117], neural tissue [118,119,120,121,122,123,124,125,126,127] and nervous system [128,129,130], heart [131,132,133,134,135,136,137,138,139], cardiomyocytes [140,141,142,143], including cardiovascular rate control [144,145], blood vessels [146,147,148,149,150], blood and formational elements [63,151,152,153,154,155,156,157,158,159,160,161], liver [162,163,164], spleen [165], pancreas [166], kidney [167], intestine [168,169]. The targets of studies include lysosomes [170,171], endoplasmic reticulum [172], membranes [173], membrane channels [174], cytoskeleton [175], recombination processes [176], aneuploidy [177], expression regulation [178], and signaling [179]. There is research on the genetic basis of the organism’s response to oxidative stress [180,181,182,183,184], hypoxia [185,186,187], sodium ion uptake from the surrounding aquatic environment and ammonia excretion [188,189,190], inflammation processes [191,192], immunity [193,194,195,196] and disease resistance [197,198,199,200,201].

The genes responsible for the economic traits of zebrafish and other fish species, such as fish growth and development, pigmentation, and sex determination, are summarized in Appendix A and are discussed in more detail below.

A number of methodological studies have been carried out on zebrafish to improve genome editing. Improving editing efficiency and minimizing side effects relative to classical gene transfer technologies are the main goals of genome editing technology. By optimizing the type and structure of Cas protein and the amount of sgRNA, researchers have developed many improved versions of genome editing technologies. The improved eSpCas9, SpCas9-HF1, and HypaCas9 were first tested on zebrafish [202,203,204]. EvoCas9 and HiFi Cas9 (Cas9 point mutation R691A), which had a low frequency of off-target bias without altering efficacy, were also tested on this species [205,206].

A protocol for the knockout of genes in zebrafish using HypaCas9 and HiFi Cas9 has been developed [207]. Scientists combined the SV40 NLS cell nucleus signaling protein with Cas12a for applications in zebrafish individuals [208]. These studies revealed that the improved Cas12a had high knockdown efficiency, and the percentages of off-target mutations and toxicity were reduced. Cas9 toxicity has been noted by many researchers [209,210,211], so the use of a less toxic Cas12a is promising. In another study, a four-guide RNAi search table for 21,386 zebrafish genes was reported using four sgRNAs and the Cas9 protein complex (four controlled anti-ribonucleoprotein (RNP) Cas9) for injection into zebrafish embryos. As a result, the knockout efficiency in the G0 generation exceeded 90%, whereas the incidence of embryo malformations was <17%. In this way, in particular, the key gene *zbtb16a* of heart development in zebrafish was identified [212].

It was shown in zebrafish that there is a correlation between chromatin accessibility and CRISPR-Cas9 mutagenesis efficiency. The results indicated that CRISPR-Cas9 mutagenesis is dependent on chromatin structure in embryos. Thus, the prokaryotic CRISPR-Cas9 system is affected by eukaryotic chromatin structures. The probability of successful mutagenesis in zebrafish embryos correlates with transcript abundancy during early development [213].

The efficiency of microinjection is one of the key factors affecting the success of gene-editing technology in fish. Microinjections into fish embryos are difficult because the eggs are sticky and the hard chorion impedes the insertion of microinjection needle. Mucus also readily adheres to the tip of the needle and causes mechanical damage to the eggs or needle breakage, so injection efficiency and embryo survival rates are often low. It has been found that 0.25% trypsin can break down the mucus covering the egg and significantly improve microinjection efficiency [214]. As development progresses, the chorion of fish eggs gradually hardens. In addition, the high internal pressure in the egg during injection can cause the contents of the egg to leak out, which reduces the probability of embryo survival. A needle with a modified tip shape can prevent the leakage of egg contents during microinjection. A needle tip sharpened at a specific angle allows easier penetration into the chorion [215]. Further research on different conditions for handling fish eggs prior to injection is needed to improve the accuracy of the instruments and the progress of microinjection. Usually, in order to avoid the mosaicism of edited fish and minimize the number of traumatic punctures of the eggs, microinjections are carried out at the stage of one- and two-cell embryos. However, modern methods allow microinjections to be carried out up to the four-cell stage [216].

The new CRISPR activation system (CRISPRa) has been tested on zebrafish. It is a convenient tool for activating target genes. It was developed and combined with an illumination-based system that can time- and localization-dependently control transcription initiation using a photoreceptor derived from the plant thale cress (*Arabidopsis thaliana*). The blue light photoreceptor cryptochrome-2 (CRY2) and its binding partner CIB1 form a dimer when exposed to blue light. Researchers activated zebrafish genes using the activators p65 and VP64 in ZF4-type zebrafish cells. The study confirmed the successful control of gene transcription levels using this system. The mRNA expression levels of the *ASCL1a*, *BCL6a*, and *HSP70* genes increased after irradiation with blue light for several hours and were significantly different from those treated in the dark [217].

Various life activities of organisms are closely related to the precise regulation of gene expression. Gene knockout provides a simple and efficient method to study gene expression and identify gene function. However, using a single genome-editing technology to solve biological problems has certain limitations. The combination of genome-editing technology and multi-omics can pave the way to a better understanding of gene function. Gene knockout can indicate the functional significance of genes identified by multi-omic analysis, and, in turn, multi-omics can characterize mutant effects at different molecular levels (such as transcription level or protein level), including the metabolic level after gene knockout. Moreover, the combination of high-throughput sequencing and gene-editing technology (mainly CRISPR/Cas9) has proven to be a good strategy for the rapid screening of functional genes on a large scale [218]. Initially, this system was applied to model organisms such as zebrafish and their cell lines [219,220].

### 2.3. Genome Editing and Body Development in Fish

Fish body development poses a crucial economic implication, particularly in relation to the formation of the fish skeleton. To enhance the marketable quality of fish products, it is advisable to reduce the number of bones within the skeletal muscle.

In zebrafish, genes such as *hoxb5b* [221], *foxc1a*, *foxc1b* [222], *bmp7a* [223], and *gne* [224] play pivotal roles in establishing symmetry along the central axis of the body. Knocking out these genes results in compromised dorsal–abdominal axis formation, abnormalities in organ arrangement, defects in visceral organ development, and embryonic mortality. Further details, including the type of editing system, mutant phenotypes, and protein annotations for the mentioned genes, can be found in Appendix A. The data presented here and below on gene functions were processed using genome editing methods. High embryo lethality may also be linked to impaired organogenesis and immunity, as evidenced by mutants in knockout genes such as LOC795232 [225], *sphk1* [226], *asap1a*, *asap1b* [227], and *zrsr2* [228]. The regulation of cartilage and bone formation, bone mineralization, and osteoblast differentiation involves *sox9* [229], *vwa1* [230], *tmem38b* [231], *ppp2r3b* [68], *col11a2* [69], and *wnt16* [232] genes. In common carp, the *sp7a* gene serves a similar function [24]. The deactivation of these genes results in severe pathologies affecting tissue formation, structure-forming cartilage and bones, collagen synthesis impairment, fin deformation, and compromised bone repair after injuries.

The genes *wnt16* [233], *hoxaa*, *hoxab*, *hoxba*, *hoxca*, *hoxda*, [234], *bmp2a*, *bmp2b* [235], and *nkx3.2* [175] control the development of the spine, skull, and fins. Mutants for these genes exhibited significant deformities in body shape. The formation of the skull and cranial body pole organs is mediated by the *mosmoa*, *mosmob* [236], *scxa*, *scxb* [237], *cyp1b1* [238], and *hspa8* [239] genes. Genes such as *stat3* [240], *kif7* [241], and *gdf5* [242] are responsible for spine development. All this work was carried out on the model object zebrafish.

To enhance the marketable quality of fish products, obtaining mutants for the knockout variant of the *runx2b* gene [243] shows promise. It controls the development of small muscle ossicles–spicules in the myoseptae (connective tissue partitions between skeletal muscle segments). These mutants lack spicules in the myoseptae, and the bone mineral density, growth, and swimming behavior of the fish remain unimpaired. Similar work was performed on Gibel carp [244].

### 2.4. Genome Editing and Growth Traits in Fish

Growth encompasses traits related to fish skeletal muscle formation, body size, glucose metabolism, thyroid hormones, insulin, leptin, polyunsaturated fatty acid synthesis, and feeding behavior. All these traits play a pivotal role in enhancing the quality of aquaculture fish products, increasing product yield, and optimizing feed amount and composition.

The control of muscle cells, skeletal muscle, and muscle contractility in zebrafish is governed by genes such as *ik* [245], *foxm1* [210,246], *dyrk1b* [246], *vcp* [247], and *tpcn1* [248]. The disruption of these genes results in a reduction in the number of muscle cells, impaired differentiation, compromised neuromuscular contacts, and deterioration of skeletal and cardiac musculature.

Knocking out genes like *katnal2* [81], *smc5* [249], *stat5.1* [250] in zebrafish and *igfbp-2b1*, *igfbp-2b2* [251,252] in rainbow trout led to the development of smaller fish compared to unedited individuals. Notable progress has been made with the knockout of myostatin-2 gene, a negative regulator of muscle tissue growth in common carp [24,253]. Mutants of the *mstnba* gene grow larger than unedited individuals, exhibiting both hyperplasia and hypertrophy of muscle tissue. Similar studies have been conducted on yellow catfish [21,254], channel catfish (*Ictalurus punctatus*) [255], olive flounder (*Paralichthys olivaceus*) [256] medaka (*Oryzias latipes*) [257], and Nile tilapia [258]. The popularity of editing this gene is not surprising, as it gives rise to a beneficial mutation that improves fish marketability. A similar effect is caused by gene *pomc* knockout [259]—an increased body weight due to more intense muscle formation without signs of obesity—and the *acvr2* gene knockout [260]—hypertrophy of muscle fibers, increased muscle growth and body weight. Both these genes are also promising for use in agriculture.

Thyroid hormones play a crucial role in stimulating the growth and development of organisms by intensifying energy metabolism. The disruption of the genes encoding these hormones by genome editing technology made it possible to clarify their function. The *greb1* gene, responsible for the formation of somatotropic, thyroid, lactotropic, and gonadotropic secretory cells in embryogenesis [261], exhibits high embryo mortality when knocked out. Silencing genes such as *tshba*, *tg*, *slc16a2* [262], *duox* [263], *tpo* [66], *isl2a*, and *isl2b* [117] also play a crucial role in thyroid hormone production. Mutants in these genes display significant defects in thyroid gland and organ development, decreased synthesis of thyroid hormones, and growth retardation.

Studies on zebrafish have explored the role of leptin, insulin, and genes regulating their function in shaping feeding behavior, maintaining glucose homeostasis, and managing energy metabolism. Two research groups found an influence of leptin *lepb* and its receptor *lepr* genes on these traits [264,265], but another group found no such influence [266]. In rainbow trout, the knockout of the leptin receptor gene resulted in a hyperphagic phenotype, increased body weight, and rapid growth [93]. Further investigations are needed to fully comprehend the role of leptin and its receptor in shaping feeding behavior and obesity. The knockout of genes such as *rfx6* [267], *akr1a1a* [268], *igf1* [269], *rreb1a*, *rreb1b* [270], and *glo2* [271] in zebrafish resulted in decreased insulin synthesis and impaired glucose homeostasis, underscoring the importance of these genes in maintaining normal sugar and energy metabolism.

Genes such as *nur77* [272], *pparγ* [273], *phlpp1* [274], and *cygb1* [275] influence fat metabolism, cholesterol, and triglyceride accumulation. Notably, the knockout of the *phlpp1* gene leads to a decrease in total cholesterol and triglyceride levels, reducing their accumulation in vessel walls. In the context of fats, genes responsible for double-chain polyunsaturated fatty acid (DC-PUFA) synthesis in zebrafish include *elovl8a*, *fads2* [276,277], and in Atlantic salmon, *fads2*, *Δ6abc/5Mt*, and *Δ6bcMt* [278,279]. Obtaining fish with a desired DC-PUFA content in fillets holds significance for aquaculture. These studies contribute to understanding the mechanism of fatty acid synthesis, particularly in response to food composition [278,279].

In zebrafish, the role of the *t1r1* gene of taste receptor type 1 in fish food behavior has been investigated [280]. Mutants with a knockout variant of this gene lose sensitivity to alanine, causing the fish to readily switch to a plant-based diet. On such a diet, the expression of the gene encoding satiety peptides increases in fish, while the synthesis of hunger peptides decreases.

### 2.5. Genome Editing Affecting Fish Pigmentation

Bony fishes exhibit diverse and colorful pigment patterns. To date, eight different pigment cell types have been identified in the skin of bony fishes, originating from the neural crest. The multipotency of neural crest cells and the diversity of pigment cells in fishes make them an ideal model for studying the formation, differentiation, and migration of different pigment cell types. Additionally, colorful ornamental fishes, such as koi carp and farm-raised fish with vibrant body colors have higher market value, prompting the application of genome-editing technologies in studies investigating fish pigmentation. Moreover, changes in pigmentation are a ready marker for the success of a genome-editing protocol, which makes these genes a convenient target for testing the method [281].

Zebrafish possess three types of pigment cells: black melanophores (accumulating melanin), orange xanthophores (accumulating carotenoids), and iridophores (containing silver and gold pigments or structures). The migration of pigment cell precursors from the neural crest to the site of differentiation is regulated by the *pcdh10a* and *pcdh10b* genes [282], which are responsible for intercellular contacts. Genes related to thyroid hormones and their receptors, such as *tg*, *scarb1*, *tyr*, *thraa*, *thrab*, and *thrb*, also play a crucial role in pigmentation formation. They direct the cell cycle of melanophores toward final differentiation and stimulate xanthophores to accumulate carotenoids [283]. Mutants for the *sox10* gene knockout variant do not develop any pigment cells, and in double mutants for *sox10* and *sox5*, normal cell differentiation is partially restored, making the *sox10* gene essential for the coloration of zebrafish. Interestingly, in the Japanese medaka, the operation of these two genes is completely opposite [284]. The formation of melanophores and xanthophores is regulated by a group of genes, including *pax3a*, *pax3b*, *pax7a*, *pax7b* [285]. Carotenoid accumulation in xanthophores is controlled by the *plin6* [286] and *scarb1* [287] genes. Proper iridophore formation is regulated by the genes *alk*, *ltk*, *alkal1* (*aug-α1*, *aug-α2*), *alkal2* (aug-β) [288], and *edn3a*, *edn3b*, and *ednrb1a* [289].

In Atlantic salmon, unpigmented mutants were obtained by deactivating the *tyr* and *slc45a2* genes [23]. Several genes responsible for melanin synthesis and melanophore distribution, including *asip1*, *asip2* [290], *mc1r* [291], and *tyrp1* [292], were identified in the colored Oujiang carp. In the goldfish, the tyrosinase gene *tyr* is responsible for melanin synthesis, similar to other fishes [293].

For the convenience of microscopy scientists, non-pigmented, completely immobile zebrafish was created by knocking out the *slc45a2* and *chrna* genes [294]. Such fish are suitable for study with light optics, as they do not require fixation and lack interfering body coloration.

Dorso-ventral counter shading, the difference in coloration between the dark dorsum and light belly, is an important characteristic of fish pigmentation. Gene knockout studies have established that the *mc1r* [295] and *asip1* [296] genes are responsible for this pigment cell distribution

### 2.6. Genome Editing and Sex Determination in Fishes

Sex determination is difficult to understand in many fish species, and its function may depend on genetics, embryogenesis, and endocrinology. Many genes have been found to co-regulate the process of sex determination. Significant dimorphism in the size of males and females is observed in at least twenty aquaculture fish species [297]. The study of sexual differentiation is thus an important research topic in aquaculture fishes with significant dimorphism. For commercial production, it is economically advantageous to raise fish of the same sex with a higher growth rate or larger size.

Zebrafish have no dimorphic sex chromosomes, but there appear to be sex-determining genes. The sex ratio is influenced by environmental factors such as oxygen content in the water, water temperature, and population density [298]. Sex is determined through the differentiation of gonads, and this process depends on the balance of steroid hormones. Steroidogenesis is a key process of hormone synthesis leading to differentiation, the development and maturation of gonads, fertility, and reproduction.

The differentiation of gonads into ovaries is determined by genes for steroid hormones and their receptors. Switching off aromatase encoded by the *cyp19a1a* gene results in knockout zebrafish forming male gonads but retaining follicles and oocyte-like cells [299,300,301,302]. Follicle formation and the activation of follicles were impaired in mutants with the knockout of the follicle-stimulating hormone *fshb* and luteinizing hormone *lhb* genes [303], their receptors *fshr* and *lhcgr* [304], and progesterone receptor *pgr* [305]. A similar effect was observed in females with *gsdf* gene knockout [306]. Accordingly, sufficient amounts of these hormones and receptors are necessary for the normal functioning of the female reproductive system. Estrogen receptors *esr2a*, *esr2b*, and *esr1* are also responsible for follicle formation and the maintenance of the female sex in adulthood. Moreover, two or three genes had to be knocked out to produce a mutant phenotype; mutants in one gene out of three did not form a mutant phenotype, suggesting that estrogen receptors are redundant in zebrafish [307]. The biosynthesis of female steroid hormones and, consequently, ovary formation and maintenance of the female sex are affected by the operation of the genes *sox3* [308], *inhbaa*, *inhbab*, *inhbb*, *bmp15*, and *inha* [309,310].

Steroid-encoding genes are also involved in male sex determination. The amount of testosterone and 11-ketotestosterone depends on the activity of hydroxylases encoded by the *cyp17a1* [311,312] and *cyp11c1* [313] genes. Interestingly, all *cyp17a1* gene knockout offspring were nevertheless males [311]. These data are in good agreement with studies of knockout of this gene in common carp [7], where only males were obtained in the F0. Such males were crossed with wild-type females; in the F1, only heterozygous females with normal ovaries and increased body weight were obtained. Obtaining such a unisex population of fish with good market qualities is promising for use in aquaculture.

Spermatogenesis and the formation of male secondary sexual characteristics in zebrafish occur under the control of androgen receptor *ar* genes [314,315,316].

The anti-Müllerian hormone can cause the regression of the Müllerian duct in mammals and is important for the differentiation of Leydig cells, as well as for the development of follicles in adult females. Zebrafish with abnormalities in the *amh* gene encoding this hormone were found to have hypertrophied gonads in both males and females, impaired germ cell differentiation due to excessive proliferation, and altered gene expression of other steroid hormones [317,318].

In addition to steroid hormones, other proteins influence sex determination and fertility. Females and males with the mutant of the *wnt4a* gene were sterile due to genital tract malformations [319]. The genes responsible for ovarian differentiation, oocyte maturation, and folliculogenesis are *myoc* [320], *egf*, *egfra*, *egfrb* [321], *nobox* [322], *parn* [323], *ambra1b* [324], *scg2a*, *scg2b* [325], *avp* [326], and *bmp15* [300,310]. Disruptions in the latter two genes affected, among other things, the mating behavior of fish. In Gibel carp, oogenesis, folliculogenesis, and gonad differentiation occurred under the control of the genes *cgfoxl2a-B*, *cgfoxl2b-A*, and *cgfoxl2b-B* [327]. The correct function of the *dmrt1* gene is critical for proper differentiation and maintenance of sperm viability in zebrafish [318,328]. The *sdY* gene is responsible for testes formation in male rainbow trout [329,330].

Genes whose knockout stops both oocyte and sperm development, namely *mettl3* [331] and *dnd* [332], were found in zebrafish. Transcripts of these genes are critical for the formation of normal gametes. The knockout of the *dnd* gene with the same result was also obtained in rainbow trout [333] and Atlantic salmon [281].

Gene knockout also was used to show the role of vitellogenin proteins (*vtg1*, *3*, *4*, *5*) of various forms in the production of mutant zebrafish. When the genes responsible for their synthesis were blocked, various phenotypic manifestations (reduced fecundity of females, impaired maturation of eggs, edema of the pericardium and yolk sac, and spinal lordosis and impaired locomotor activity in larvae), as well as compensatory mechanisms not previously observed, were observed. These results provided evidence that different types of *vtg* proteins in vertebrates fulfill different essential functions during both reproduction and embryonic development. Interestingly, *vtg1* mutants had an increased number of eggs, although the larvae died due to multiple developmental defects during late developmental stages [334]. Defective eggs with disrupted pineal structures in the shell are formed in mutants of the *stm* gene [335].

### 2.7. Gene Editing and Disease Resistance in Fishes

Viral and bacterial infections of fish are often responsible for significant economic losses in aquaculture. The study of disease-resistance factors in fish is an emerging area of agricultural genetics. The use of conventional breeding methods for this purpose has a number of limitations. For example, inbreeding effects accumulate rapidly in small populations of disease-resistant animals. Such populations can become a reservoir for newly emerging infections. Unfortunately, under these conditions, there is also a significant possibility that when resistance to a target pathogen emerges, susceptibility to another pathogen will develop. In addition, specimens selected for breeding may be carriers of a pathogen, such as a virus, and transmit it to their progeny [336]. Gene editing promises to overcome these limitations.

Recently, works investigating the function of genes involved in disease resistance by knockout using the CRISPR/Cas system have begun to appear. Of the species under consideration, such works have been performed only on the model object zebrafish.

A series of works is devoted to the study the functions of genes involved in the immune response reaction. The *gpr56* gene coordinates the expression of other immune response genes and digestive enzyme genes [193]. The *socs3b*, *mitfa*, *CD18*, and *lcp1* genes control the morphology, number, or migration of neutrophils and macrophages to the site of inflammation [154,159,194,195]. Knockout mutants of the *fam76b* gene produce hyperinflammation in response to antigen activity [337]. The *stat5.1* gene controls T-cell lymphopoiesis and fish growth [196]. Switching off all these genes leads to impaired immune response, defects of immune cell development, an increase in their number, impaired functionality, incorrect migration when non-target organs are infiltrated, and inadequately bright inflammation in response to pathogen activity.

The role of two zebrafish genes in resistance to viral diseases was studied using the CRISPR/Cas9 method: the *viperin* gene in relation to the viral hemorrhagic septicemia virus [197] and the *rrm1* gene in relation to the nervous necrosis virus [338]. In both cases, mutants showed greater susceptibility to viruses than wild-type fish. Hence, both of these genes determine resistance to their respective diseases. The *prmt7* gene determines susceptibility to spring viraemia of carp virus (SVCV) and grass carp reovirus (GCRV) [339]. The knockout of it is promising for use in aquaculture. In Asian sea bass (*Lates calcarifer*), the *gab3* susceptibility gene for nervous necrosis virus was detected. *gab3* plays an important role in virus replication, and its suppression leads to lower virus titers in infected fish and increased survival during infection [340]. A susceptibility gene for grass carp reovirus has been found in grass carp [25]. In the grass carp kidney cell line with *gcJAM-A* gene knockout, the cytopathogenic effect of the virus was reduced. The *gab3* genes in Asian sea bass and *gcJAM-A* in grass carp are candidates for genomic editing.

For some bacterial infections, resistance genes have also been found in zebrafish. And knockout of them is not desirable. The *sting* [199] and *tnf-α1* [341] genes are responsible for resistance to *Edwardsiella piscicida*, *crp* for resistance to *Streptococcus pneumoniae* [200], and *cxc3.3* for resistance to *Mycobacterium* spp. [342]. At the same time, the *cxc3.2* gene determines susceptibility to these bacteria [342]. The *itln3* gene is neutral with respect to mycobacteria [62]. It is likely that the specificity of the above resistance genes to bacterial infections is broader than to a single bacterial species. Studies are needed to verify this. In channel catfish, the myostatin gene was found to mediate susceptibility to *Edwardsiella ictalurid* [343]. The knockout of this gene in aquaculture fish may be of interest.

## 3. Overcoming Technical Difficulties in Applying Genetic Engineering Approaches in Aquaculture

Genome editing requires a clear and reliable knowledge of the genome of a species; however, data on the genomes of aquatic species are still scarce, although many of the genomes of the most important aquatic animals have been sequenced [344]. Since the study of the first sequenced fish species tiger puffer (*Takifugu rubripes*) in 2002, which is a significant achievement in recent decades [345], the genomes of more than one hundred fish species have been decoded due to advances in sequencing technology and a relative decrease in sequencing costs. However, the number of such species is still small compared to the total number of species used in aquaculture, which according to FAO exceeds six hundred [1]. Therefore, the application of CRISPR/Cas in aquaculture would benefit from further refinement of databases.

Another important challenge for scientists is to study the functional role of genes associated with important phenotypic traits. Since the genetic study of aquatic organisms lags behind that of humans and plants, it is necessary to identify genes associated with specific traits. It is also important that the further development of genome-editing technologies requires a reduction in the cost of sequencing (preferably to less than ten dollars per sample). This will lead to more species-specific genomes of aquatic animals being decoded in the future, which in turn will provide the necessary information base for future studies on the application of genomic editing [13].

The process of identifying target genes through quantitative trait locus (QTL) mapping or using genetic markers is a time-consuming process. Although re-sequencing technology is now facilitating it, a polygenically determined trait still makes it difficult to accurately identify candidate genes [13]. Improvements in molecular biology methods (e.g., QTL mapping, genome-wide association studies (GWASs) dedicated to detecting variants at genomic loci that are associated with complex traits in the population, comparative genomics, and pooled CRISPR screens, which are a synthesis of large-scale libraries of sgRNA for functional genetic screens) will lead to the identification of more genes associated with the traits under study [346]. On the other hand, individual specific alleles responsible for favorable traits for species and lines should also be considered as candidates for refining genome-editing techniques.

Chromosomal duplication in fish also creates difficulties in working with their genomes. Among aquatic vertebrates, fishes represent the taxonomic group with the largest number of species. Many bony fish species are characterized by whole-genome duplications [347]. The mechanism by which genome-editing efficiency is reduced in polyploid fishes is currently under debate [251,279,348,349]. To clarify this issue, a comparison of different copies of genes within the genome is required for the further efficient application of genomic editing.

It should also be noted that the shell of the egg reduces microinjection success. Despite advances in technology, due to the different biological characteristics of fishes, there is no single standardized protocol for them with a universal needle type, injection dosage, etc. For ovoviviparous fishes, no single platform for gene editing has been developed at present either [13].

Another problem with genome editing is the presence of off-target effects when editing loci that are not targets of manipulation but contain a sequence similar to that of the target gene to which the gRNA of the editing complex binds. Options for preventing or detecting off-target mutations include careful design of annealing RNA by comparison with existing genome assemblies or by screening for unplanned mutations after editing. Regarding the latter, natural genetic variability across families and lineages makes off-target mutations difficult to detect after editing [9].

When working with traits involving the expression of several genes, the simultaneous creation of multigenic knockout mutants using CRISPR/Cas provides the possibility of inducing the desired phenotype. Great technical successes have been achieved for some fish species with the production of various genetically modified (GM) lines, especially Atlantic salmon and Nile tilapia. These species should be used as “aquaculture models” for optimization of CRISPR/Cas protocols and potential off-target effects (for food quality and safety assessment). Knowledge gained from this approach can also be utilized in other fish species [13].

In several aquaculture species, the intergenerational interval is quite long (they tend not to spawn every year) or the full lifecycle cannot be completed in captivity, which makes the detection of edited individuals with homozygous genotype during genome editing very laborious. A possible solution to the problem is to combine genome editing with surrogacy technology to reduce the generation interval or marine phase by using the xenogenic transplantation of germ cells [350].

Individuals that are not capable of producing offspring are thought to be most suitable for maintenance after genome editing for two main reasons. First, the maturation of the gonads consumes a large amount of resources and body energy, which negatively affects the quality of the final aquaculture product. Also, mature animals become more susceptible to disease and stress [351]. Second, when escaping into the environment, sterile animals will not interbreed with wild populations or affect their genetic diversity, and their progeny will not displace wild species from their habitats, even if aquaculture animals have an ecological advantage [352,353]. However, it requires additional effort to maintain heterozygous individuals to sustain the population.

There are several strategies for obtaining sterile fish [354,355]. The first technologies were medical—gonadectomy, exposure to high doses of radiation, feeding or incubation in media with high concentrations of synthetic androgens, and hyperthermia [355]. These have subsequently been supplanted by genetic technologies.

The most common of these is triploidization. For example, triploid Atlantic salmon are produced by applying high hydrostatic pressure to newly fertilized eggs, which leads to the preservation of the second polar body in the egg division, the non-disjunction of chromosomes, and the preservation of two sets of chromosomes from the mother and one from the father in the embryo [356]. Triploids can also be obtained by exposing the cell to colchicine or cytochalasin B at the time of division [357]. In rainbow trout, triploids are obtained by crossing tetraploids and diploids [358]. Triploid salmon are inferior to diploid salmon in terms of their properties: they are smaller in size and poorer in marketability, and they survive in artificial breeding conditions [359]. It may be difficult to use triploids in aquaculture in the future [355].

Sterile offspring also occur in some cases in interspecific hybrids. Such hybrids have a potential heterosis effect, showing improved traits inherited from both parents. However, fish hybrids have not shown increased productivity, and many have been shown to be fertile [360,361,362,363].

There are systems that induce silencing by antisense RNAs of sterility-inducing genes with the possibility of their subsequent activation. The gonadotropin-releasing hormone gene *GnRH* is often used as a target [364]. Silencing of the dead-end *dnd* gene with morpholino [365] or Gal4/UAS system [366,367] has also been developed. Potentially, the induced expression of suicide genes or toxalbumin, Diphtheria toxin in fish germinal cells could be used to generate sterility. A disadvantage of this approach is that toxins can accumulate in the body of the fish and then be released into the environment or be potentially harmful to the consumer [354]. Another unsafe method is the use of the Ntr/Met system, which is based on the conversion of metronidazole to cytotoxin using the enzyme nitroreductase. The expression of nitroreductase is controlled by a gonad-specific promoter, and metronidazole is supplied to cultivation water [368,369]. Finally, a nifty tet-on/off system for reversible sterility induction has been developed wherein fish are sterile under normal conditions and become fertile in the presence of tetracycline [370].

The above methods of sterility formation in fish require their genetic modification or transgenesis. This goal can also be achieved by genomic editing. Candidate genes include *wnt4a*, which leads to sterility in both males and females due to defects in reproductive tract development [319], *mettl3* [331], and *dnd* [332], whose knockout mutants do not form germ cells. Homozygous females with *pgr* gene knockout and *lhb* [303] and male double mutants for the *fshr* and *lhcgr* genes [304] are also sterile.

## 4. Conclusions

Genome editing technology is applicable to a large number of fish species and can contribute not only to basic research but also to the realization of useful traits in farmed fish species [371].

An important task in improving economically valuable traits of fish using genome editing is the choice of the target gene. In an ideal situation, it should influence one discrete trait, cause a minimal off-target effect, and form the desired mutant phenotype. While the first two problems can be more or less solved bioinformatically or using modern improved CRISPR/Cas systems, it is still difficult to predict the mutant phenotype. Appendix A allows us to navigate the variety of work already completed in the gene editing of Salmonidae and Cyprinidae species and facilitate target gene selection.

Fish with knockouts of the following genes may be the most promising for use in aquaculture. Work on common carp has shown that mutants of the myostatin gene *mstnba* have large bodies with increased muscle cell number [24,253]. A similar effect is caused by the knockout of the genes *pomc* [259] and *acvr2* [260] in zebrafish. Experiments on zebrafish and Gibel carp allow fish to be obtained without smaller intramuscular bones in the myoseptae of muscles, the formation of which is driven by the *runx2b* gene [243]. The disruption of the leptin receptor gene *lepr* allows the formation of a hyperphagic phenotype, leading to the accelerated growth of fish. [304,306]. The technologies under consideration make it possible to obtain Atlantic salmon with a programmable content of DC-PUFAs in the meat [278,279]. Producing large heterozygous females from the offspring of *cyp17a1* knockout fathers and wild-type mothers is promising [7] for common carp. Also of interest are absolutely sterile mutants of the *dnd* gene. Such experiments have been performed on zebrafish [332], rainbow trout [333], and Atlantic salmon [281]. Pioneering work to restore germ cells in *dnd* knockout mutants was performed [372]. In order to achieve this paradoxical goal, the investigation introduced full-length stabilized mRNA of the *dnd* gene into embryos at the time of editing. The restored *dnd* gene had mutations, but immature gametes were formed in such fish. By developing this approach, it is possible to obtain *dnd*-/- mutants without crossing heterozygous parents but directly from genetically sterile fish. Studies using rainbow trout revealed that *dnd* mutants without their own gametes can act as recipients of gametes from wild relatives [333], providing a surrogate parent useful for conservation applications. Sterile fish should be obtained by knockout *mettl3* [331] and *wnt4a* [319] genes. Investigations were performed on zebrafish.

Work on zebrafish has shown that fish with reduced pigmentation and motility with the *slc45a2* and *chrna1* genes turned off can be useful for laboratory experiments [294].

Also of interest is the knockout of disease susceptibility genes—*prmt7* in relation to Spring viraemia of carp virus and Grass carp reovirus [339], work was performed on zebrafish; *gab3* in relation to Nervous necrosis virus [336], work was performed on Asian sea bass; *gcJAM-A* to Grass carp reovirus [25] in grass carp, *mstnb* to *Edwardsiella ictalurid* [343] in channel catfish, and *cxc3.2* to *Mycobacterium* spp. [342] in zebrafish. 

## Figures and Tables

**Figure 1 genes-15-00726-f001:**
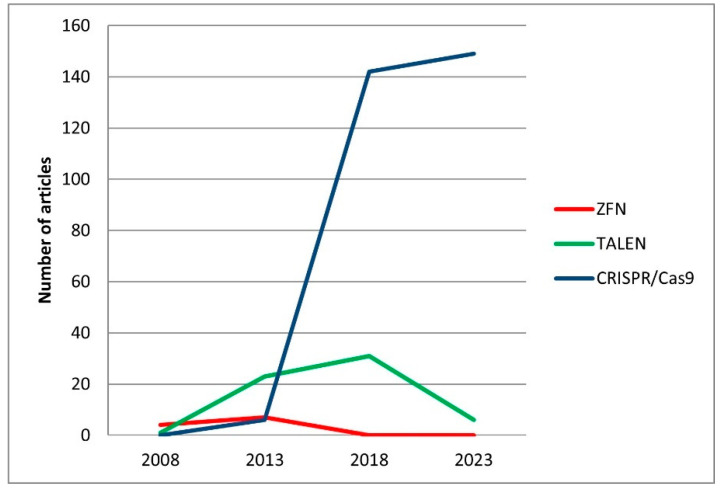
Number of articles on zebrafish genome editing in the PubMed article database from 2008 to 2023 using ZFN, TALEN, and CRISPR/Cas9 technologies.

## Data Availability

No new data were created or analyzed in this study. Data sharing is not applicable to this article.

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
