# Peer review of "In Search of a Target Gene for a Desirable Phenotype in Aquaculture: Genome Editing of Cyprinidae and Salmonidae Species"

_genes, 2024, doi:10.3390/genes15060726_

Round 1
Reviewer 1 Report
Comments and Suggestions for Authors
Genes-3004487
This manuscript explained major genome editing methods, reviewed comprehensive studies related to genome editing using fish species and cited 475 references. The authors better to focus on Cyprinidae and Salmonidae species as in the title of this manuscript. And nice to have a list of genes potentially contribute to the aquaculture based on references discussed in this review, mainly from zebrafish.
Major comment
1. Lines 78-354: Delete basic information of genome editing methods. No need this part, since most of scientists using CRISPER/Cas9. If authors believe that genome editing technologies other than CRIPER/Cas9, please highlight the advantage(s) of them with reason.
2. Lines 414-453: this part is explanation of disease model of zebrafish, therefore, this part can be deleted or cite book chapters and/or some important reviews. Or disease related to fish can be kept.
3. Scientific names should be added once after English names, thereafter, only use English names
Minor editorial revisions
1. Lines 20-21: rainbow trout (Onchorhynchus mykiss), Atlantic salmon (Salmo salar), common carp (Cyprinus carpio), gold fish (Carassius auratus), silver carp (C. gibelio) and the model fish zebrafish (Danio rerio)
2. Line 43: add space before [2]
3. Lines 51-53, 66: add parentheses for the scientific names of fish)
4. Lines 72-74: delete scientific names
5. Line 366 and throughout the text: yellow catfish (Tachysurus fulvdraco) (2011) add parentheses for the scientific name, delete scientific names which already appeared in the text
6. Line 384: delete Danio rerio
7. Line 411: specify the four different species
8. Line 547: Danio rerio should be zebrafish
9. Line 584: also play a
10. Line 635: add Japanese before medaka
11. Line 709: add space before [430,431]
12. Lines 739, 750: Danio rerio should be zebrafish
13. Line 776: add comma after however
14. Line 841: add space before [454]
15. Line 916: obtain should be obtained
16. Line 923: add space after [445]
17. References should be followed the journal style: titles should not be large capitals, journal names should be appropriately abbreviated and italic, scientific names of animals should be italic]
Author Response
This manuscript explained major genome editing methods, reviewed comprehensive studies related to genome editing using fish species and cited 475 references. The authors better to focus on Cyprinidae and Salmonidae species as in the title of this manuscript. And nice to have a list of genes potentially contribute to the aquaculture based on references discussed in this review, mainly from zebrafish.
Dear reviewer, thank you for your comments on our work, which helped to make the manuscript more relevant and structured.
Major comment
- Lines 78-354: Delete basic information of genome editing methods. No need this part, since most of scientists using CRISPER/Cas9. If authors believe that genome editing technologies other than CRIPER/Cas9, please highlight the advantage(s) of them with reason.
Done, we agree with this comment and have these sections removed.
- Lines 414-453: this part is explanation of disease model of zebrafish, therefore, this part can be deleted or cite book chapters and/or some important reviews. Or disease related to fish can be kept.
Thank you for the comment, we would like to save the part with the list of diseases, because we believe that this part illustrate an ability of genome editing to deal with both monogenic and complex genetic diseases, therefore to elucidate and uncover functional genomics in fish.
- Scientific names should be added once after English names, thereafter, only use English names
Thank you for the comment, we have fixed it.
Minor editorial revisions
- Lines 20-21: rainbow trout (Onchorhynchus mykiss), Atlantic salmon (Salmo salar), common carp (Cyprinus carpio), gold fish (Carassius auratus), silver carp ( gibelio) and the model fish zebrafish (Danio rerio)
Thank you for comment; we added English names of the species.
- Line 43: add space before [2]
Thank you, done.
- Lines 51-53, 66: add parentheses for the scientific names of fish)
Thank you, done.
- Lines 72-74: delete scientific names
Thank you, done.
- Line 366 and throughout the text: yellow catfish (Tachysurus fulvdraco) (2011) add parentheses for the scientific name, delete scientific names which already appeared in the text
Thank you, done.
- Line 384: delete Danio rerio
Thank you, done.
- Line 411: specify the four different species
Thank you, done.
- Line 547: Danio rerio should be zebrafish
Thank you, done.
- Line 584: also play a
Thank you, done.
- Line 635: add Japanese before medaka
Thank you, done.
- Line 709: add space before [430,431]
Thank you, done.
- Lines 739, 750: Danio rerio should be zebrafish
Thank you, done.
- Line 776: add comma after however
Thank you, done.
- Line 841: add space before [454]
Thank you, done.
- Line 916: obtain should be obtained
Thank you, done.
- Line 923: add space after [445]
Thank you, done.
- References should be followed the journal style: titles should not be large capitals, journal names should be appropriately abbreviated and italic, scientific names of animals should be italic]
Thank you for the comment, now we have references section fixed.
On behalf of authors,
Dr. Nikolai S. Mugue
Reviewer 2 Report
Comments and Suggestions for Authors
Overall, this is a well written and comprehensive review. I have only a few minor comments which may help enhance the manuscript. The edits consist of a few additional references to consider.
Line 239: need a reference here for the cytosine over abundance. See Pribylova et al 2022 PMID35524464. This one is on plants but there's probably a similar one on fish?
Line 260... finally resolved.... I think a different word from "resolved" would be a better descriptor for this sentence.
Line 307. Just a comment but there are a few other examples of knockouts that are beneficial. See Mokrani and Liu 2024 Aquaculture
Line 518: need a reference here. I realize that there are references at the end of the paragraph but you should insert one here to break it up.
Line 624: need a general reference for this statement.
Author Response
Overall, this is a well written and comprehensive review. I have only a few minor comments which may help enhance the manuscript. The edits consist of a few additional references to consider.
Dear reviewer, thank you for your attention to our work and reviewing! We also thank you for your comments to make the article more accurate and informative.
Line 239: need a reference here for the cytosine over abundance. See Pribylova et al 2022 PMID35524464. This one is on plants but there's probably a similar one on fish?
Thank you for very interesting suggestion, we believe that this epigenetic modifications on the efficacy of CRISPR/Cas9-mediated double-stranded DNA breaks and subsequent DNA repair is poorly understood and should be considered when dealing with fish as well. However, in accordance with the recommendations of the other reviewer, we have removed parts of MS that dealt with general technical aspects of genome editing (that are widely covered in the literature elsewhere). This exact paragraph (former section 2 of the original MS) has been removed in the revised version.
Line 260... finally resolved.... I think a different word from "resolved" would be a better descriptor for this sentence.
This paragraph has also been removed.
Line 307. Just a comment but there are a few other examples of knockouts that are beneficial. See Mokrani and Liu 2024 Aquaculture
Thank you for your note, we have mentioned Mokrani and Liu 2024 in the appropriate sections.
Line 518: need a reference here. I realize that there are references at the end of the paragraph but you should insert one here to break it up.
Thank you, we have added Huang et al., 2022 as well.
Line 624: need a general reference for this statement.
Thank you, we added Wargelius et al. 2016 to support importance of pigmentation-related gene as a proxy of successful editing of the target gene.
On behalf of authors,
Dr. Nikolai S. Mugue
Round 2
Reviewer 1 Report
Comments and Suggestions for Authors
The authors adequately revised the manuscript following comments from the reviewer. No further comments on this revised manuscript.